# Diabetic Keratopathy: Redox Signaling Pathways and Therapeutic Prospects

**DOI:** 10.3390/antiox13010120

**Published:** 2024-01-18

**Authors:** Francesco Buonfiglio, Joanna Wasielica-Poslednik, Norbert Pfeiffer, Adrian Gericke

**Affiliations:** Department of Ophthalmology, University Medical Center, Johannes Gutenberg University Mainz, Langenbeckstrasse 1, 55131 Mainz, Germany; joanna.wasielica-poslednik@unimedizin-mainz.de (J.W.-P.); norbert.pfeiffer@unimedizin-mainz.de (N.P.)

**Keywords:** diabetic complication, cornea, keratopathy, redox, pathways, molecular, targets

## Abstract

Diabetes mellitus, the most prevalent endocrine disorder, not only impacts the retina but also significantly involves the ocular surface. Diabetes contributes to the development of dry eye disease and induces morphological and functional corneal alterations, particularly affecting nerves and epithelial cells. These changes manifest as epithelial defects, reduced sensitivity, and delayed wound healing, collectively encapsulated in the context of diabetic keratopathy. In advanced stages of this condition, the progression to corneal ulcers and scarring further unfolds, eventually leading to corneal opacities. This critical complication hampers vision and carries the potential for irreversible visual loss. The primary objective of this review article is to offer a comprehensive overview of the pathomechanisms underlying diabetic keratopathy. Emphasis is placed on exploring the redox molecular pathways responsible for the aberrant structural changes observed in the cornea and tear film during diabetes. Additionally, we provide insights into the latest experimental findings concerning potential treatments targeting oxidative stress. This endeavor aims to enhance our understanding of the intricate interplay between diabetes and ocular complications, offering valuable perspectives for future therapeutic interventions.

## 1. Introduction

Diabetes mellitus (DM) stands out as the most prevalent endocrine disorder, characterized by chronic hyperglycemia. A disruption in insulin secretion from the pancreatic beta cells, impaired insulin action, or a combination of both are its cause. As of 2019, global DM prevalence was 8.3%, projected to rise to 9.6% by 2045 [1]. Correspondingly, the projected surge in global DM costs is staggering, soaring from $1.3 trillion in 2015 to a projected $2.1 trillion by 2030 [2]. While diabetic retinopathy and cataracts are recognized ophthalmological manifestations, the correlation between DM and eye surface disorders is less acknowledged. This may stem from patients reporting eye surface symptoms primarily in advanced stages. DM ranks as a leading risk factor for dry eye disease (DED) development [3]. The duration of DM correlates with DED manifestation [4], and higher glycated hemoglobin (HbA1c) levels elevate the likelihood of DED, with a critical level of HbA1c reported to be 8–8.5% for men and 7.7–7.8% for women [5,6,7]. Furthermore, a link between diabetic retinopathy severity and DED intensity has been noted [8]. Altered tear film composition, reduced production, and secretion contribute to tear film instability [9,10]. Patients with DM face increased risks of corneal changes, including superficial punctate keratitis, recurrent erosions, persistent epithelial defects, and ulcers. In addition, a recent meta-analysis indicated that DM may exert detrimental effect on corneal endothelial cells [11]. These diabetic corneal changes are termed diabetic keratopathy, were initially identified as a distinct entity by Schultz et al. in 1981, and affect up to 70% of all individuals with DM [12,13,14,15,16,17]. Hence, diabetic keratopathy emerges as a relatively common complication of DM, and in advanced stages, it poses a serious threat to vision [18]. Unfortunately, in clinical practice, this disorder is often underdiagnosed and overlooked [19].

Corneal alterations in diabetes manifest through increased corneal thickness, epithelial defects, fragility, superficial punctate keratitis, delayed and incomplete wound repair, as well as detectable endothelial changes and neuropathy characterized by decreased corneal sensitivity [20]. In fact, peripheral diabetic neuropathy contributes to diabetic keratopathy by damaging corneal nerve fibers due to hyperglycemia. This results in a significant reduction and alteration in subepithelial nerve fiber morphology, leading to reduced corneal surface sensitivity, a main characteristic of diabetic keratopathy [21,22]. Progressive corneal epithelial cell loss, recurrent erosions, ulcers, and scarring events may culminate in devastating corneal opacities, ultimately causing severe visual impairment or blindness [23,24]. Figure 1 illustrates an example of diabetic keratopathy.

## 2. Pathophysiology of Diabetic Keratopathy

### 2.1. Corneal Cell Damage in Diabetic Keratopathy

#### 2.1.1. Redox Signaling in the Diabetic Cornea

Hyperglycemia, the primary contributor to the pathogenesis of DM and its complications, notably affects the corneal epithelium, a crucial tissue involved in diabetic keratopathy. The corneal epithelium, being a major player in this condition, exhibits insulin-independent glucose uptake through glucose transporter-1 (GLUT1), facilitating a continuous influx of glucose into cells [25,26,27]. In instances of heightened metabolic demand, such as during corneal damage, the expression of GLUT1 transporters increases [28]. Furthermore, the corneal epithelium directly receives glucose through transcorneal transport from the aqueous humor, rather than from tears, thereby maintaining high glycogen levels. During DM, the corneal epithelium becomes exposed to high glucose levels [29]. Notably, hyperglycemia triggers an escalation in the production of advanced glycation end products (AGEs) and significantly elevates the levels of ROS [30,31,32,33,34]. Studies by Kim and colleagues revealed that in diabetic corneal cells, the accumulation of AGEs is linked to oxidative DNA damage, measured through 8′-hydroxy-2′-deoxyguanosine levels. This cascade leads to apoptotic activation via the nuclear factor kappa-light-chain-enhancer of activated B cells (NF-kB), contributing to delayed epithelial wound healing in the diabetic cornea [35]. AGEs, generated under hyperglycemic conditions, promote ROS generation by binding to their receptor RAGE (receptor of AGE). This process stimulates enhanced nicotinamide adenine dinucleotide phosphate (NADPH) oxidase (NOX)-related ROS formation and increased phosphorylation of p47^phox^. This, in turn, activates the c-Jun N-terminal kinase and p38 mitogen-activated protein kinase pathways, culminating in apoptosis and microvascular complications [36,37,38,39]. Additionally, hyperglycemia increases intracellular concentrations of diacylglycerol (DAG), activating protein kinase C (PKC) and subsequently NOX subtypes (NOX1, NOX2, and NOX4), collectively amplifying ROS formation [38,40,41,42,43]. Furthermore, high glucose levels inhibit the epidermal growth factor receptor (EGFR)–phosphatidylinositol 3-kinase (PI3K)/Ak strain transforming (Akt, also known as protein kinase B) pathway through ROS, attenuating corneal epithelial wound healing [44].

As an additional repercussion of the oxidative stress condition, NF-kB, a crucial transcription factor that regulates immune responses and the expression of adhesion molecules, becomes activated. This activation leads to the abnormal production of proinflammatory cytokines, contributing to chronic inflammation. This establishes a positive loop between oxidative stress, inflammation, and apoptosis in diabetic keratopathy [35]. In addition, ROS and RAGE trigger phospholipase C/PKC/extracellular signal-regulated kinase activator protein 1 signaling, promoting fibrogenesis through transforming growth factor beta (TGF-β) expression [45]. Notably, the TGF-β1 isoform, a major subtype expressed in the cornea, induces delayed re-epithelialization and myofibroblast transformation, suggesting its inhibition as a potential strategy to promote corneal wound healing [46]. Of note, RAGE signaling is activated not only by AGEs but also by other ligands, such as the chemokine high-mobility group box 1 protein (HMGB1), whose expression is reported to be elevated in the corneas of diabetic murine models during wound healing. Its inhibition may be a potential molecular target [47].

#### 2.1.2. Pyroptosis in the Corneal Epithelium Triggered by Hyperglycemia

ROS abundance not only triggers aberrant inflammation via NF-kB but also induces epithelial cell death via pyroptosis. Oxidative stress can activate the nucleotide-binding domain, leucine-rich-containing family, pyrin domain-containing-3 (NLRP3) inflammasome in corneal epithelial cells [48]. The NLRP3 inflammasome is a cellular multiprotein complex that plays a significant role in the immune response. The protein complex is activated when cells are damaged or infected by certain stress factors such as environmental toxins, bacteria, or viruses [49,50,51]. The NLRP3 inflammasome consists of the NLRP3 protein responsible for detecting harmful signals, the apoptosis-associated speck-like protein containing a caspase activation and recruitment domain (ASC), and the caspase-1 protein responsible for the activation of pro-inflammatory cytokines like interleukin (IL)-1ß and IL-18 [43,52]. When the NLRP3 inflammasome is activated, caspase-1 provides cleavage and activation of IL-1β and IL-18, as well as of gasdermin D, whose fragments generate a membrane pore, allowing proinflammatory cytokine release and membrane disruption, finally leading to pyroptosis [53]. These pathological processes lead to cell death in epithelia and stromal keratocytes.

#### 2.1.3. Involvement of Stroma, Endothelium, and Tight Junctions in Diabetic Damage

Bitirgen et al. demonstrated a reduced basal epithelial cell density, decreased anterior stromal keratocyte counts, and endothelial cell density occurring prior to diabetic retinopathy in individuals with DM type 2 [54]. Zhang and colleagues reported a specific susceptibility of corneal endothelial cells to ultraviolet A-related oxidative damage in a diabetic murine model. This was attributed to an attenuated parkinsonism-associated deglycase 1/nuclear factor erythroid-derived 2-related factor 2 (Nrf2)/nicotinamide adenine dinucleotide (NAD) phosphate (P)H quinone oxidoreductase 1 (NQO1) pathway [55]. Nrf2 activation is closely related to the levels of nuclear deacetylase sirtuin 1 (SIRT1) [56], with an upregulation of SIRT1 associated with increased Nrf2 activation [57]. Both proteins are essential antioxidative regulators, suggesting their significance as molecular targets in antioxidant research [58].

Additionally, hyperglycemia has been reported to impair conjunctival and corneal barrier function, potentially through a loss of laminin-5, a principal component of the corneal basement membrane [59]. Reduced expression of the tight junction component occludin was observed in the human diabetic corneal epithelium by Huang and colleagues [60]. Furthermore, hyperglycemia delays corneal wound healing and reduces cell migration via oxidative stress, through impairing Akt signaling [44,61,62]. More specifically, the phosphatase and tensin homologue protein results upregulated in damaged corneal epithelium, and its inhibition may allow an increase in the PI3K/Akt signaling, accelerating wound healing [63,64].

Figure 2 illustrates the mechanisms leading to a loss of corneal epithelial cells in diabetic keratopathy. The increased loss of corneal epithelial cells can lead to a reduction in the barrier function of the corneal epithelium, thereby favoring superficial punctate keratitis, recurrent erosions, persistent epithelial defects, and infections [65].

### 2.2. Diabetic Corneal Neuropathy

#### 2.2.1. Evidence of Corneal Nerve Damage in Diabetes

The cornea represents the most densely innervated structure in the human body, with its nerves contributing to various homeostatic functions such as blinking movements and the release of neuropeptides and growth factors [12,66,67]. The central corneal nerve density is approximately 7000 nociceptors/mm^2^, making the cornea roughly 300–600 times more sensitive than the skin [68]. Stromal nerve branches are formed by 900–1200 myelinated and unmyelinated axons with a diameter ranging between 0.5 and 5 μm [22]. The peripheral and central nerve endings of the human cornea consist of nociceptive Aδ and C fibers [69]. Stromal nerves are disposed parallelly to the collagen lamellae and subdivide into smaller fascicles as they extend in superficial/anterior direction, creating interconnections within the anterior stromal plexus [67,70]. While some stromal nerves terminate as free endings, a significant number of them directly innervate the keratocytes [70,71], establishing a profound and intricate relationship between corneal epithelial layers and sensory innervation [72], particularly pertinent in cases of DM. The stroma-originated nerves give rise to the sub-basal nerve plexus (SBNP), which innervates the corneal epithelium. SBNP fibers pass through the Bowman’s layer into the epithelium, predominantly in peripheral regions. These fibers form long bundles progressing towards the center, dividing into smaller branches that interconnect, ultimately creating a complex network within the epithelium [66]. Figure 3 illustrates the corneal layers and innervation, emphasizing the distinctive distribution and ramification of stromal nerves throughout the epithelium.

Multiple studies have elucidated the intricate mechanisms of neuroimmune cross-talk within the cornea, primarily mediated through the interplay between neuromodulators expressed by corneal nerves and their receptors in resident corneal leukocytes [73,74]. Corneal sensory innervation releases neurotrophins and neuropeptides, while autonomic innervation secretes catecholamines and acetylcholine [21,68]. Neurotrophic molecules play a crucial role in preserving corneal homeostasis and facilitating corneal wound healing [75]. Notably, neurotrophin-3 (NT-3) and nerve growth factor (NGF) are pivotal neurotrophic agents for ensuring corneal health and repair [76].

The sensory nerves within the cornea produce various neuropeptides responsible for triggering diverse molecular pathways. Among these are substance P, calcitonin gene-related peptide, and α-melanocyte-stimulating hormone, which activate different molecular signaling pathways [77]. Additionally, beyond conventional neurotransmitters, the sympathetic innervation in the cornea also involves serotonin and neuropeptide Y. Meanwhile, as regards the corneal parasympathetic innervation, activity was demonstrated for vasoactive intestinal polypeptide and galanin [77]. Neuropeptides exert their effects by binding to receptors belonging to the superfamily of G protein-coupled receptors, thereby initiating various downstream signaling pathways [77]. Of particular interest in managing diabetic keratopathy might be substances like substance P [78], possibly in combination with insulin-like growth factor 1 [79,80], and vasoactive intestinal polypeptide [81,82]. These substances play roles in promoting wound healing and corneal regeneration. Delayed wound healing, reduced corneal sensitivity, and anomalies in the tear film associated with DM are intricately linked to profound injuries to corneal nerve fibers during the early phases of DM [22,83,84,85]. This correlation underscores an imbalance in the interdependence between corneal epithelium and nerves, leading to damages in both tissues, characteristic of DM-related neurotrophic keratopathy [86,87]. Numerous investigations have detailed critical morphological changes and a pathological decrease in SBNP fibers in DM, even during prediabetes [88,89,90]. Specifically, diabetic patients exhibit reductions in nerve fiber density, branch density, and fiber length [91,92,93]. Structural changes induced by DM extend to the highly innervated stromal cornea, where the accumulation of AGEs contributes to collagen crosslinking and increased corneal thickness [94]. Moreover, abnormal increases in the tortuosity of stromal nerves have been observed in diabetic patients [95]. Overexpression of matrix metalloproteinases (MMP)-3 and -10 in the diabetic stroma indicates significant corneal alterations and remodeling [96]. Rodent diabetic models demonstrated stromal inflammation and edema [97], and in diabetic murine models, dendritic cell infiltration and nerve fiber impairment in the corneal SBNP hint at a neurotrophic function of this cell subpopulation [98]. Moreover, corneal sub-epithelial microneuromas and axonal swelling positively correlate with painful diabetic polyneuropathy, underlining the nuanced relationship between corneal nerve morphology and diabetic complications [99].

In both experimental and clinical studies, the most significant decrease in corneal branch density and nerve fibers occurs in the SBNP, near the epithelium, emphasizing the strong association between corneal neuropathy and diabetic keratopathy [100,101,102,103,104,105]. Reduced corneal keratocytes in diabetic patients have been linked to damage in SBNP fibers [106]. An increase in corneal Langerhans cells is associated with decreased SBNP fiber density [107], while diminished keratocyte density and increased Langerhans cell density have been linked to delayed epithelial wound healing [108]. Furthermore, the variability in serum glucose levels and the duration of DM play crucial roles in the detrimental effects on corneal nerves. For example, individuals with DM under insulin-dependent therapy for five or more years displayed diminished corneal epithelial nerve density and an abundance of nerve loops in the stroma [101]. Longer durations of DM type 2 are positively correlated with the gradual degeneration of SBNP [109]. Moreover, a recent study highlighted morphological changes in corneal stromal nerves in DM, underscoring an association between glucose level variability and the loss of corneal nerve length in the inferior whorl [110].

Collectively, the reduced SBNP fiber density, increased nervous tortuosity, and decreased epithelial nerve fiber length with fewer branches, when compared to controls, are strongly associated with attenuated corneal sensitivity in DM [22,95,100,111,112,113]. These changes reflect a deficiency in the normal neurotrophic connection between corneal innervation and the epithelium, contributing to corneal epithelial defects, erosions, infections, ulcers, and stromal opacifications [86,114].

#### 2.2.2. Molecular Pathways Leading to Corneal Nerve Damage

From a molecular standpoint, hyperglycemia triggers the activation of several pathways involving mitochondria, AGEs, the polyol pathway, and the PKC cascade. These pathways collectively contribute to an excess of ROS, ultimately leading to cell death and corneal nerve injuries [21,90].

##### Hyperglycemia-Related Mitochondrial Dysfunction

In the context of DM, the influx of glucose into the mitochondria accelerates the electron transport chain, indirectly promoting the increased formation of superoxide (O_2_^●−^), resulting in oxidative stress [30]. Hyperglycemia induces alterations in mitochondrial dynamics, with short-term exposure leading to fission, caspase-3 activation, and oxidative stress, while prolonged exposure may trigger mitochondrial biogenesis [115]. Chronic hyperglycemia contributes to impaired mitochondrial axonal transport due to the accumulation of larger, swollen mitochondria, resulting in decreased nerve conduction velocity, axonal demyelination, and neuron apoptosis [116,117,118]. Importantly, these mitochondrial anomalies can disrupt the corneal neurotrophic interaction by decreasing the levels of NT-3 and NGF [76,119].

##### AGEs and Axonal Degeneration

During chronic hyperglycemia, AGEs not only overproduce ROS but also form cross-links with structural proteins in peripheral nerves. This process induces the formation of glycated myelin proteins, dysfunctional entities that promote demyelination, impaired regenerative activity, and axonal degeneration [120,121,122]. AGEs also cross-link with collagen molecules in the corneal stroma [94], leading to the glycation of laminin and fibronectin in Descemet’s membrane. This glycation decreases the adhesion of corneal endothelial cells [123], potentially causing endothelial dysfunction and microangiopathy [21].

##### Effects of the Polyol Pathway and PKC Cascade

The polyol pathway converts glucose into sorbitol and then into fructose, and its activation is a result of hyperglycemia-induced overload in the glycolytic pathway. This alternative metabolic pathway necessitates NADPH and NAD^+^, leading to a decrease in glutathione (GSH) synthesis and subsequently hydrogen peroxide (H_2_O_2_) generation. As NADPH is vital for the regeneration of the antioxidant GSH, its reduced availability results in a diminished antioxidant capacity [124]. Furthermore, an abundance of sorbitol and fructose is negatively correlated with free nerve myoinositol, a molecule crucial for normal nerve conduction velocity [125]. The reduced bioavailability of myoinositol leads to decreased functionality of the membrane sodium–potassium adenosine triphosphatase (Na^+^/K^+^ ATPase), compromising nerve conduction velocity [29].

As described above, the PKC pathway is activated in response to high glucose-induced intracellular DAG, causing enhanced ROS generation through NOX activation in corneal epithelial cells [38,40,41,42,43]. Additionally, PKC activation can impair Na^+^/K^+^ ATPase function, resulting in reduced nerve conduction and regeneration [126,127].

Figure 4 summarizes the main molecular pathways associated with hyperglycemia that contribute to corneal nerve damage, along with the principal changes in the corneal stroma and epithelium, focusing particularly on the SBNP during diabetic keratopathy.

### 2.3. Diabetes-Associated Dry Eye Disease Promotes Oxidatives Stress and Inflammation

The tear function unit, encompassing cornea, conjunctiva, lacrimal gland, Meibomian glands, eyelids, and sensory/motor nerve fibers, intricately regulates tear fluid secretion and tear film formation, which are crucial for maintaining and safeguarding the tear film [128]. Disruptions in any component of this unit can result in DED. This condition is not only linked to a deficiency in tear production, but also associated with a status of chronic inflammation and oxidative stress. In this regard, a recent innovative study by Yang and colleagues demonstrated the effectiveness of a gabapentin/nanoceria nanoformulation as a carrier with anti-inflammatory, antiangiogenic, antiapoptotic, and neuroprotective properties in a rabbit model of DED [129].

Regarding DM, chronic hyperglycemia can potentially harm nerve fibers that supply the lacrimal gland, resulting in histological alterations within the lacrimal gland itself [130]. DM-induced corneal nerve fiber loss diminishes the corneal sensitivity, subsequently impairing tear secretion [12]. Moreover, patients with DM often exhibit a reduced blink frequency, a common factor in evaporative DED [3,131]. Conjunctival squamous metaplasia and a decline in conjunctival goblet cells contribute to decreased mucin secretion, diminishing tear film adhesion to the ocular surface and causing tear film instability [132]. Elevated levels of insulin-like growth factor-binding protein 3 (IGFBP3) in diabetic tears act as a negative modulator of IGF-1 signaling in the corneal epithelium, exacerbating corneal impairment and DED in diabetic corneas [133]. Furthermore, increased concentrations of AGEs in the tear fluid initiate inflammatory reactions and oxidative stress on the ocular surface, further impairing corneal epithelial cell function [134,135]. Notably, HMGB1, an alternative ligand of the receptor for RAGE, and dual oxidase 2 (DUOX2), a subtype of NOX, play crucial roles in DED. Hyperosmolarity-induced DUOX2 activation, mediated by toll-like receptor 4 (TLR4)-dependent signaling and exacerbated by ROS excess, leads to HMGB1 release, cell death, and inflammation in a positive loop in human corneal epithelial cells exposed to DED conditions [136]. In a diabetic murine model, hyperglycemia induces a more severe mitochondrial bioenergetic deficiency in lacrimal gland cells compared to corneal cells. This deficiency results in elevated mtDNA damage, reduced mtDNA copies, decreased levels of GSH, contributing to the early onset of DM-induced DED [137].

Collectively, diabetic alterations affecting tear film composition, secretion, and stability manifest as reduced tear volume and decreased break-up time and conjunctival goblet cell density. There are enhanced tear film osmolarity, squamous conjunctival metaplasia, disturbances in the ocular surface microbiota, heightening the risk of bacterial conjunctivitis, and impairment of Meibomian glands [138].

Figure 5 illustrates the tear film sublayers and the detrimental impact of hyperglycemia and ROS on tear production and stability, leading to DED.

## 3. Innovative Antioxidative Approaches for Diabetic Keratopathy

Herein, we delve into experimental preclinical findings exploring the efficacy of antioxidative molecules for managing diabetic keratopathy, with the goal of expediting corneal wound healing, alleviating diabetic DED, and mitigating diabetic corneal nerve damage.

### 3.1. Targeting Pyroptosis in Diabetic Keratopathy

Derivatives of vitamin A and D have been tested for the treatment of diabetic keratopathy in various preclinical investigations. For example, in an in vivo study on a rat diabetic model, orally administered β-carotene displayed hypoglycemic and antioxidant effects, improving diabetic keratopathy [139]. The antioxidant activity of vitamin D compounds is debated [140,141,142]. However, calcitriol, a vitamin D derivative, demonstrated in an in vitro investigation on high glucose-exposed human corneal epithelial cells, inhibitory effects on the ROS/NLRP3/IL-1β axis, enhancing corneal wound healing and innervation in diabetic mice [143]. In an in vivo study in a mouse model of DM, topical calcitriol administration was shown to ameliorate corneal wound healing and innervation, via the suppression of the NLRP3 inflammasome [144]. Moreover, in another in vivo study on diabetic mice, derivatives such as 1,25-dihydroxyvitamin D (1,25 Vit D) and 24,25-dihydroxyvitamin D (24,25 Vit D) administrated topically accelerated corneal wound healing [145].

Another molecule exerting beneficial effects via the inhibition of the ROS/NLRP3/caspase-1/IL-1β pathway is N-acetylcysteine, a well-known antioxidant, used for the treatment of paracetamol overdose and of respiratory diseases. In an in vivo study on diabetic mice, topically administered N-acetylcysteine markedly reduced ROS formation and ocular surface damage [146].

α-Lipoic acid is a naturally occurring molecule with recognized antioxidative features, which attenuated AGE accumulation, blocking AGE–RAGE–ROS–mediated oxidative stress and cells apoptosis, and suppressed AGE–RAGE–TLR4–NLRP3 cascade-related inflammation in an in vitro study on high glucose-exposed human corneal epithelial cells [147].

In patients with diabetic DED, eye drops based on a combination of α-lipoic acid and hydroxy-propyl-methylcellulose were shown to improve the effectiveness of the therapy and self-regeneration, improving corneal defects [148]. Intriguingly, a combination of α-lipoic acid with Eudragit E100 as a carrier molecule enhanced corneal permeation, and thereby α-lipoic acid delivery, favoring its retention in the cornea [149].

### 3.2. Targeting Nrf2 and SIRT1 in Diabetic Keratopathy

Glycyrrhizin is a natural antioxidant with anti-inflammatory and antifibrotic activity, which activates Nrf2 [150]. In an in vitro, in vivo, and ex vivo study on a diabetic murine model, oral doses of glycyrrhizin attenuated corneal inflammation and oxidative stress and reduced, among others, the levels of HMGB1, IL-1β, TLR2, TLR4, and NLRP3 [151]. In another in vivo investigation in diabetic mice, the subconjunctival injection of glycyrrhizin mitigated RAGE and TLR4 molecular pathway activation, thereby promoting corneal epithelial wound healing [47].

Dimethyl fumarate, already licensed for the management of multiple sclerosis and psoriasis, is a well-known compound with antioxidant and anti-inflammatory properties that showed an effective immunosuppressive action after corneal allograft transplantation in rats, blocking lymphangiogenesis and inhibiting inflammation through the suppression of NF-kB [152]. Moreover, a derivate of dimethyl fumarate, VP13/126, was shown in an in vitro study on glucose-impaired rabbit corneal epithelial cells to activate the Nrf2/heme oxygenase 1 (HO-1) pathway, inducing corneal re-epithelialization [153].

SIRT1 was demonstrated to facilitate angiogenesis in diabetic wounds by protecting endothelial cells from oxidative stress, suggesting its potential role in the treatment of diabetic skin ulcers [154]. SIRT1 is also crucial in the modulation of insulin secretion and sensitivity, further favoring cell survival by blocking apoptosis by deacetylation of p53. It was shown that the upregulation of SIRT1 promoted diabetic corneal epithelial wound healing by inhibiting p53 and activating the IGF-1R/Akt cascade [61]. Furthermore, in vivo investigations on diabetic mice based on subconjunctival injections of specific microRNAs able to modulate the activity of SIRT1, demonstrated that microRNA-182, a SIRT1 downstream effector, upregulated SIRT1 and downregulated NOX4, leading to diabetic corneal nerve regeneration [155], whereas blockade of microRNA-204-5p accelerated corneal epithelial wound healing via the upregulation of SIRT1 [156].

In a recent in vitro and in vivo investigation testing eye drop administration in a DED murine model, salidroside, a main component of the plant *Rhodiola crenulate*, was demonstrated to attenuate oxidative stress in DED, activating Nrf2 via an adenosine monophosphate-activated protein kinase (AMPK)–SIRT1 signaling pathway [157]. Rosiglitazone, a peroxisome proliferator activator receptor-gamma (PPARγ) agonist, was recently shown to decrease oxidative stress in the lacrimal gland, in part by activating PPARγ and thus promoting the overexpression of antioxidant enzymes such as glutathione peroxidase 3 (GPx3) in a diabetes-related DED murine model receiving the compound via oral gavage [158]. Quercetin, a natural occurring antioxidant administered to diabetic mice in the diet, was demonstrated to improve tear function through the upregulation of SOD1 and SOD2 in the lacrimal gland. In the same study, quercetin decreased ROS formation and enhanced cell survival in murine corneal cell lines [159].

Mitoquinone is a well-known mitochondria-targeted antioxidant [160], which was reported to activate the Nrf2/HO-1 axis in high glucose-induced brain microvascular endothelial cells, protecting them from mitochondrial ROS-related apoptosis [161]. Fink et al. reported in an in vivo study that the introduction of mitoquinone to the diet of obese and diabetic rats ameliorated motor and sensory nerve conduction velocity, corneal and intraepidermal nerve fiber density, as well as corneal sensitivity and thermal nociception [162].

Nicotinamide mononucleotide enhanced cell viability and restored tight junctions in an in vitro study on high glucose-treated human corneal epithelial cells through the activation of the SIRT1/Nrf2/HO-1 cascade [163]. Moreover, in an in vivo study in diabetic mice, topically administered DNase I promoted corneal epithelial wound healing and nerve regeneration by activating Akt, IGFR-1, and SIRT1, while inhibiting NOX2 and NOX4 upregulation and reducing ROS formation [164].

### 3.3. Counteracting Oxidative Stress and Inflammation

Pycnogenol, a standardized extract from the French maritime pine bark, *Pinus pinaster*, is a compound with photoprotective, anti-inflammatory, and antioxidant properties [165] that was reported to accelerate corneal wound re-epithelialization in an in vivo study in a diabetic rat model via eye drop administration [166]. In an in vitro study on human corneal epithelial cells exposed to oxidative stress, thymosin beta-4, a naturally occurring polypeptide, was reported to upregulate antioxidative enzymes such as superoxide dismutase (SOD) [167], reducing corneal inflammation and improving corneal re-epithelialization in ocular surface diseases [168]. An in vivo study reported that a topical application of the mitochondria-targeted antioxidant SkQ1 ameliorated the severity of DED and diabetic keratopathy in mice through improved mitochondrial function [137]. In an in vivo study in a rat model of glycotoxicity, oral administration of cemtirestat, an aldose reductase inhibitor and antioxidant molecule, was reported to reduce oxidative stress and inflammation by diminishing the levels of tumor necrosis factor alpha (TNF-α), IL-1β, NF-kB, and caspase-3 in the cornea, retina, sclera, and lens, further improving the GSH/GSSG ratio and the activity of glutathione S-transferase [169].

It is worth noting that topical insulin has been explored and has shown promising outcomes in the management of diabetic keratopathy [170]. Mechanistically, insulin has the ability to counteract the downregulation of the PI3K/Akt axis linked to ROS. This action triggers the signaling pathway, resulting in an antiapoptotic effect that promotes cell proliferation and migration, consequently expediting corneal wound healing, as evidenced by in vitro studies on human and canine corneal epithelial cells [171]. Additionally, an in vivo study in diabetic mice revealed that topical insulin can prompt corneal nerve repair by activating the Wnt/β-catenin pathway [172]. As extensively reviewed by Leong et al., both preclinical investigations and clinical trials have evaluated the efficacy of topical insulin in treating corneal defects in diabetic patients, showing promising results [173]. For example, Fai et al. demonstrated that a dose regimen of 0.5 units/drop administered four times per day proved most effective in repairing corneal epithelial defects in diabetic patients following vitrectomy, outperforming both placebo and higher concentrations [174]. Other trials displayed the effectiveness of topical insulin also in different doses, like 1.0 unit/drop four times per day, for example to treat diabetic DED [175], or better improvement of re-epithelialization in comparison with autologous serum to manage persistent corneal defects [176]. Collectively, preclinical and clinical studies have affirmed the safe and effective profile of topical insulin in managing diabetic keratopathy. This treatment accelerates corneal wound healing, potentially leading to improved outcomes for patients. 

Topical application of the pigment epithelium-derived factor ameliorated corneal epithelial injury, enhanced corneal sensitivity, and increased tear volume in a murine model of DM. The mechanism of action includes mitigation of ROS generation, reduction in RAGE expression, and upregulation of SOD-1 [177].

Recombinant human fibroblast growth factor-21, in an in vitro and in vivo study, promoted corneal epithelial wound healing by reducing the levels of pro-inflammatory markers, including TNF-α, IL-6, IL-1β, the monocyte chemoattractant protein-1, interferon gamma, MMP-2, and MMP-9, and enhancing those of antioxidative enzymes, such as SOD-1, in both diabetic mouse corneal epithelium and human corneal epithelial cells treated with high glucose [178].

In summary, according to the existing literature, the most promising antioxidants are vitamin D derivatives, α-lipoic acid, and Nrf2 activators such as glycyrrhizin, dimethyl fumarate, and mitoquinone. However, a number of other compounds have been tested and are being examined. One limitation of this review is that we included only studies written in English. Therefore, some therapeutic approaches may have remained unrecognized in this article. Table 1 provides a summarizing overview of the experimental molecules showing promise in counteracting diabetic keratopathy, with a focus on their chemical structure, mechanism of action, and molecular targets.

## 4. Conclusions

Diabetic keratopathy, a debilitating ocular complication affecting over half of the diabetic population, poses a substantial risk of corneal opacification and irreversible visual impairment. This issue underscores the critical need for innovative therapeutic interventions to address the significant challenges faced by patients and society.

This review synthesizes the current molecular insights into the processes contributing to diabetic cornea and DED. The goal is to pave the way for the design and evaluation of novel treatment strategies. Specifically, we shed light on the intricate redox pathomechanisms impacting corneal epithelium, corneal innervation, and tear film stability. Preclinical investigations exploring antioxidant compounds for the management of diabetic keratopathy showed promising results, demonstrating enhancements in corneal wound healing, nerve regeneration, and improved functionality of the lacrimal gland.

Crucially, DM can affect various sublayers and components of the cornea, including the epithelium, the stroma, the endothelium, the lacrimal gland, and the tear film. This diversity results in specific pathogenetic effects and clinical consequences within the anterior eye segment. Consequently, a multitargeted therapy designed to counteract different molecular pathways becomes feasible. Antioxidative molecules, when employed as supplementary medications alongside established drugs, have shown potential in addressing the multifaceted nature of diabetic corneal alterations. Notably, there is limited information on the effectiveness of the described compounds for human use, since most of the reported approaches were tested in preclinical investigations in cell cultures or experimental animals. Furthermore, only few studies tested also carrier molecules such as Eudragit E100 for an improved delivery of the drug into the cornea. Hence, the effectiveness and safety of corneal antioxidants for human use need thorough validation in future studies. It is imperative to leverage the insights gained from preclinical findings and translate them into well-designed clinical trials. Such trials hold the promise of delivering substantial benefits to patient outcomes, contributing significantly to the ongoing battle against diabetic corneal alterations and DED.

## Figures and Tables

**Figure 1 antioxidants-13-00120-f001:**
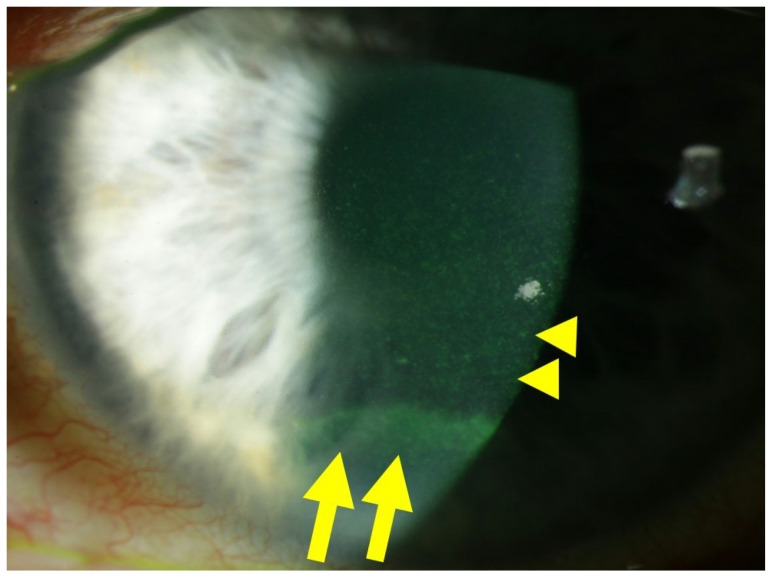
Slit lamp photograph of a patient with diabetic keratopathy, suffering from dry eye symptoms and recurrent corneal erosions. Fluorescein staining of the ocular surface reveals fine punctate keratitis (yellow dots), which is typical of dry eye disease (DED), highlighted by yellow arrowheads. In the inferior third of the cornea, an epithelial ridge (yellow horizontal line) is visible, indicative of a recently closed erosion, highlighted by yellow arrows.

**Figure 2 antioxidants-13-00120-f002:**
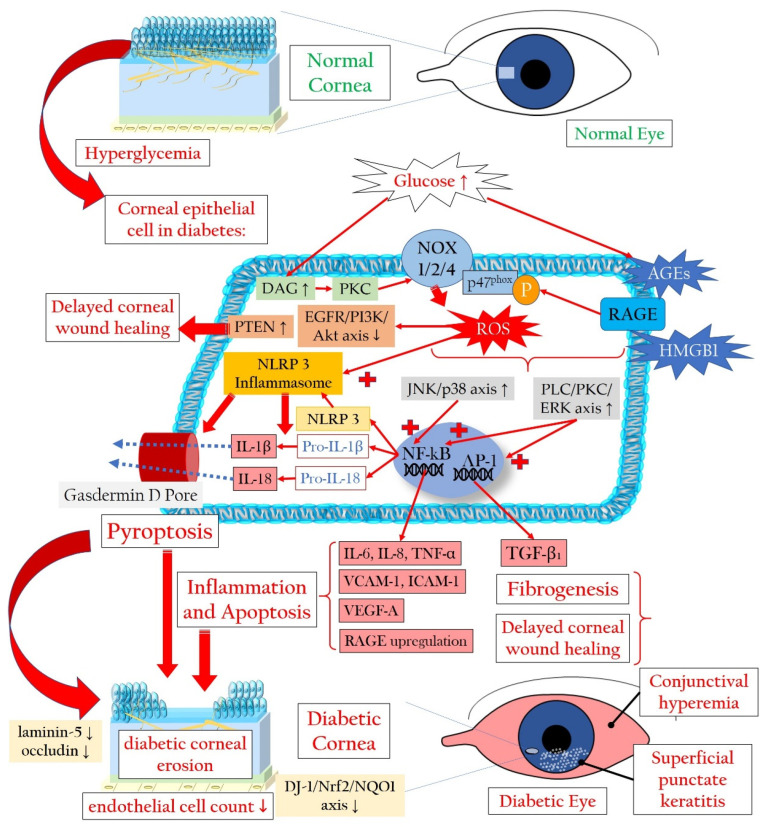
Cellular signaling mechanisms that lead to the demise of corneal epithelial cells in the context of diabetic keratopathy. AGEs: advanced glycation end products; Akt: Ak strain transforming (also known as protein kinase B); DAG: diacylglycerol; IL: interleukin; DJ-1: parkinsonism-associated deglycase 1; EGFR: epidermal growth factor receptor; ERK: extracellular signal-regulated kinase; HMGB1: high-mobility group box 1 protein; ICAM-1: intercellular adhesion molecule 1; NF-κB: nuclear factor kappa-light-chain-enhancer of activated B cells; NOX: NADPH oxidase or nicotinamide adenine dinucleotide phosphate oxidase; NQO1: NAD(P)H quinone oxidoreductase 1; Nrf2: nuclear factor erythroid-derived 2-related factor 2; P-p47phox: phosphorylated form of the protein p47phox, part of the NADPH oxidase enzyme system; PKC: protein kinase C; PI3K: phosphatidylinositol 3-kinase; PLC: phospholipase C; PTEN: phosphatase and tensin homologue protein; RAGE: receptor for advanced glycation end products; ROS: reactive oxygen species; TGF-β_1_: transforming growth factor beta 1; TNF-α: tumor necrosis factor alpha; VCAM-1: vascular cell adhesion molecule 1; VEGF-A: vascular endothelial growth factor A. Upward arrows indicate upregulation or increased activity, downward arrows indicate downregulation or decreased activity.

**Figure 3 antioxidants-13-00120-f003:**
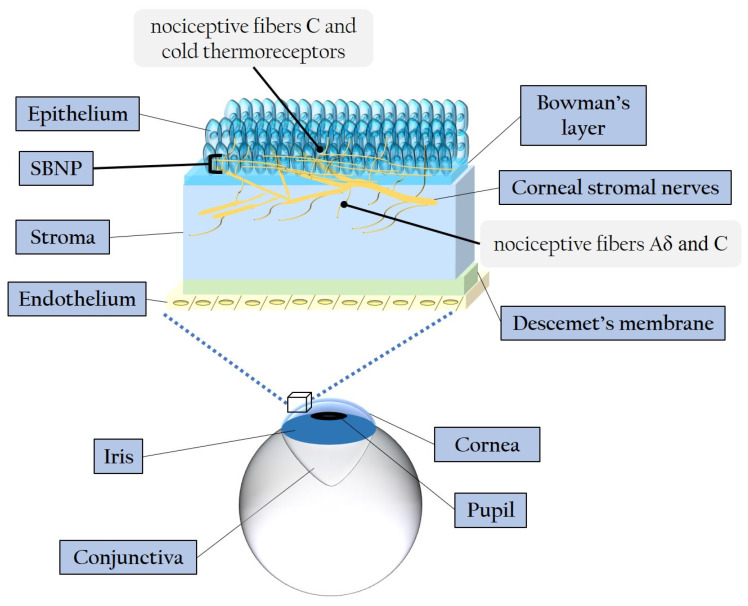
Schematic representation of the corneal layers and innervation. SBNP: sub-basal nerve plexus.

**Figure 4 antioxidants-13-00120-f004:**
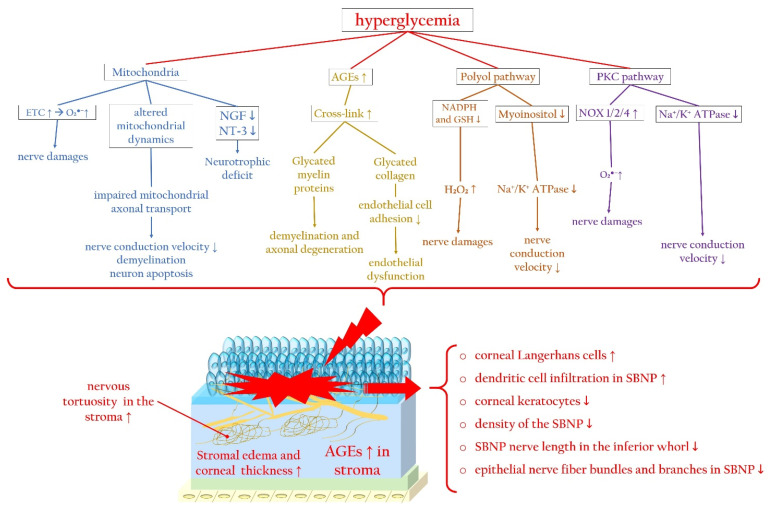
Hyperglycemia-related molecular pathways initiating corneal nerve damage and changes in the corneal stroma and SBNP during diabetic keratopathy. Hyperglycemia causes mitochondrial dysfunction, altered mitochondrial dynamics, and neurotrophic deficits, which trigger ROS overproduction. In addition, during diabetes, the accumulation of AGEs promotes the glycation of myelin, favoring axonal degeneration, and collagen, inducing endothelial dysfunction. Furthermore, activation of the polyol and PKC signaling pathways trigger, via ROS overproduction, corneal nerve damage, impairing nerve conduction. AGEs: advanced glycated end products; ETC: electron transport chain; Na^+^/K^+^ ATPase: sodium–potassium adenosine triphosphatase; NGF: nerve growth factor; NOX: nicotinamide adenine dinucleotide phosphate oxidase; NT-3: neurotrophin-3; PKC: protein kinase C; SBNP: sub-basal nerve plexus. Upward arrows indicate upregulation or increased activity, downward arrows indicate downregulation or decreased activity.

**Figure 5 antioxidants-13-00120-f005:**
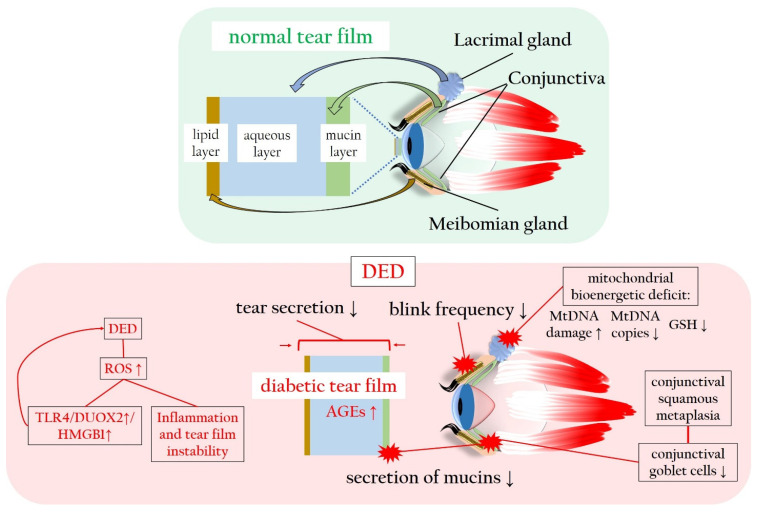
Normal and diabetic tear film in comparison, with particular focus on the ROS-related changes in tear film production, composition and stability. AGEs: advanced glycated end products; DED: dry eye disease; DUOX2: dual oxidase 2; GSH: glutathione; HMGB1: high-mobility group box 1 protein; mtDNA: mitochondrial DNA; RAGE: receptor of advanced glycated end products; ROS: reactive oxygen species; TLR4: toll-like receptor 4. Upward arrows indicate upregulation or increased activity, downward arrows indicate downregulation or decreased activity.

**Table 1 antioxidants-13-00120-t001:** Overview of experimental compounds showing promise in improving diabetic keratopathy.

Molecule	Chemistry/Biologic Effect	Study Design, Route, Model	Molecular Target and Study Outcome	Ref.
β-carotene	vitamin A derivative	in vivo, oral application, diabetic rat model	antioxidant and hypoglycemic effect, ameliorating corneal changes	[139]
calcitriol	vitamin D derivative	in vitro, high glucose-treated human corneal epithelial cells	inhibition of ROS–NLRP3–IL-1β signaling via activation of Nrf2 antioxidant signaling	[143]
		in vivo, topical administration, diabetic mouse model	promotion of diabetic corneal wound healing and reinnervation via NLRP3 suppression	[144]
		in vivo, topical route with 1,25 Vit. D or 24,25 Vit. D, diabetic mouse model	improvement of corneal wound healing	[145]
NAC	N-acetylated derivative of the natural amino acid L-cysteine	in vivo, topical administration, diabetic mouse model	mitigation of ocular surface damage via suppression of the ROS/NLRP3/caspase-1/IL-1β signaling pathway	[146]
ALA	(R)-enantiomer of lipoic acid: vitamin-like fatty acid	in vitro, high glucose-exposed human corneal epithelial cells	suppression of AGE–RAGE–TLR4-NLRP3 pathway-induced inflammation and amelioration of oxidative stress, apoptosis, and inflammation	[147]
		Eye drops based on a combination of ALA and HPMC in diabetic patients with DED	effectiveness in the treatment of diabetic DED and self-regeneration, improving corneal defects	[148]
GLY	naturally occurring saponin	in vitro, in vivo, ex vivo, oral application, diabetic mouse model	downregulation, among others, of HMGB1, IL-1β, TLR2, TLR4, and NLRP3, leading to attenuation of corneal inflammation and oxidative stress	[151]
		in vivo, subconjunctival injection, diabetic mouse model	attenuated activation of RAGE and TLR4 molecular pathways, promoting corneal epithelial wound healing	[47]
VP13/126	DMF derivative	in vitro, glucose-impaired rabbit corneal epithelial cells	activation of the Nrf2/HO-1 pathway, inducing corneal re-epithelialization	[153]
SIRT1 modulators	miRNA	in vivo, subconjunctival injection, diabetic mouse model	miRNA-182 upregulates SIRT1 and downregulates NOX4, promoting diabetic corneal nerve regeneration	[155]
		in vivo, subconjunctival injection, diabetic murine model	blockade of microRNA-204-5p favors corneal epithelial wound healing via SIRT1	[156]
salidroside	glycoside, extract from *Rhodiola crenulata*, natural antioxidant	in vitro and in vivo, eye drops, DED murine model	mitigation of oxidative stress in DED through Nrf2 via AMPK–SIRT1 signaling on the ocular surface	[157]
rosiglitazone	thiazolidinedione, insulin-sensitizing drug	in vivo, oral gavage, diabetes-related DED in a mouse model	decrease in oxidative stress in the lacrimal gland in part by activating PPARγ, inducing overexpression of antioxidants such as GPx3	[158]
quercetin	flavonol, naturally occurring antioxidant	in vivo, diet route, diabetic mouse model	improvement of tear function in diabetic mice via upregulation of SOD1 and SOD2 in the lacrimal gland, reduction of ROS formation, and promotion of cell survival	[159]
mito-Q	synthetic drug, mitochondria-specific antioxidant	in vivo, diet route, diet-induced obese or type 2 diabetic rat models	amelioration of nerve conduction velocity, corneal and intraepidermal nerve fiber density, corneal sensitivity, and thermal nociception	[162]
NMN	nucleotide	in vitro, high glucose-treated human corneal epithelial cells	enhancement of cell viability by reducing apoptosis, increasing cell migration, and restoring tight junctions via activation of the SIRT1/Nrf2/HO-1 axis	[163]
DNAse I	enzyme responsible for DNA degradation	in vivo, topical administration, diabetic mouse model	improvement of corneal epithelial wound healing and nerve regeneration by activating Akt, IGFR-1, SIRT1, while inhibiting NOX2 and NOX4 upregulation, reducing ROS	[164]
pycnogenol	mixture of flavonoids and procyanidins	in vivo, eye drops, diabetic rat model	acceleration of wound re-epithelialization	[166]
thymosin β-4	naturally occurring polypeptide	in vitro, human corneal epithelial cells exposed to oxidative stress	upregulation of antioxidants such as SOD	[167]
SkQ1	mitochondria-targeted antioxidant	in vivo, topical administration, diabetic mouse model	amelioration of DED severity and diabetic keratopathy via improvement of mitochondrial function	[137]
cemtirestat	aldose reductase inhibitor and antioxidant	in vivo, oral administration, rodent model for glycotoxicity	reduction of inflammation and oxidative stress via TNF-α, IL-1β, NF-kB, and caspase-3 downregulation	[169]
insulin	growth factor with regenerative and antiapoptotic effects	in vitro, human and canine corneal epithelial cells	activation of the PI3K/Akt axis, leading to antiapoptotic effects, favoring cell proliferation and migration, accelerating corneal wound healing	[171]
		in vivo, eye drops, diabetic murine model	enhancement of the corneal nerve repair via activation of the Wnt/β-catenin pathway	[172]
		insulin eye drops in diabetic patients with diverse dose regimens (0.5 or 1.0 unit/drop, 2–4 times daily)	enhancement of corneal epithelial wound healing, mitigation of diabetic DED, improvement of re-epithelialization compared with autologous serum in persistent corneal defects	[174,175,176]
PEDF	growth factor with antioxidant effects	in vivo, topical administration, diabetic mouse model	reduction of corneal epithelial defects via mitigation of ROS generation, decreased RAGE expression, and upregulation of SOD-1	[177]
rhFGF-21	growth factor with anti-inflammatory and antioxidant properties	in vitro on human corneal epithelial cells and in vivo on a diabetic mouse model	promotes corneal epithelial wound healing by reducing pro-inflammatory markers like TNF-α, IL-6, IL-1β and promoting antioxidant enzyme expression such as that of SOD-1	[178]

Table Abbreviations. AGEs: advanced glycated end products; Akt: Ak strain transforming (also known as protein kinase B); ALA: α-lipoic acid; AMPK: adenosine monophosphate-activated protein kinase; DED: dry eye disease; DMF: dimethyl fumarate; GLY: glycyrrhizin; GPx3: glutathione peroxidase 3; HMGB1: high-mobility group protein B1; HO-1: heme oxygenase-1; HPMC: hydroxypropyl methylcellulose; IGFR-1: insulin-like growth factor receptor 1; IL-1β: interleukin beta 1; miRNA: micro RNA; Mito-Q: mitoquinone; NAC: N-acetylcysteine; NF-kB: nuclear factor ‘kappa-light-chain-enhancer’ of activated B cells; NLRP3: nucleotide-binding domain, leucine-rich-containing family, pyrin domain-containing-3; NMN: nicotinamide mononucleotide; NOX: nicotinamide adenine dinucleotide phosphate oxidase; Nrf-2: nuclear factor erythroid-derived 2-related factor 2; PEDF: pigment epithelium-derived factor; PPARγ: peroxisome proliferator-activated receptor gamma; RAGE: receptor of advanced glycated end products; rhFGF-21: recombinant human fibroblast growth factor-21; ROS: reactive oxygen species; SIRT1: sirtuin 1; SOD: superoxide dismutase; TLR: toll-like receptor; TNF-α: tumor necrosis factor alpha.

## Data Availability

Not applicable.

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
