# Peer review of "Diabetic Keratopathy: Redox Signaling Pathways and Therapeutic Prospects"

_antioxidants, 2024, doi:10.3390/antiox13010120_

Round 1
Reviewer 1 Report
Comments and Suggestions for Authors
The current review aims to summarize the redox signaling pathways and therapeutic prospects for diabetic keratopathy. Although the topic is interesting in its scientific field, there are some issues that require the authors’ attention to improve the quality of this particular manuscript before further consideration for publication in a high-quality journal “Antioxidants”.
Specific comments:
1. Figure 1 shows slit lamp photograph of a patient with diabetic keratopathy, suffering from dry eye symptoms and recurrent corneal erosions. Fluorescein staining of the ocular surface reveals fine punctate keratitis (yellow dots), which is typical for dry eye disease (DED). In the inferior third of the cornea, an epithelial ridge (yellow horizontal line) is visible, indicative of a recently closed erosion. However, the yellow dots and horizontal line are unclear to the audiences. The authors should add some labelling markers to imaging data to identify these important features.
2. In Section “3. Innovative Antioxidative Approaches for Diabetic Keratopathy”, the authors delve into experimental preclinical findings exploring the efficacy of antioxidative molecules for managing diabetic keratopathy, with the goal of expediting corneal wound healing, alleviating diabetic DED, and mitigating diabetic corneal nerve damage. Nevertheless, in order to allow easy reading and understanding, this reviewer strongly suggests the inclusion of tabular data by summarizing the findings from these literature reports.
3. As stated by the authors, the disruptions in any component of this unit can result in DED. However, this important claim was not supported by any documented reference. If possible, please refer to the following example paper (DOI: 10.1021/acsnano.3c07817) involving the therapeutic strategy for effective alleviation of DED. The authors are highly recommended to consider the inclusion of this supportive case study in the reference list to update article content and balance scientific viewpoint.
Author Response
We thank the reviewer for the comments and suggestions. According to these suggestions, we made changes in the text.
Specific comments:
- Figure 1 shows slit lamp photograph of a patient with diabetic keratopathy, suffering from dry eye symptoms and recurrent corneal erosions. Fluorescein staining of the ocular surface reveals fine punctate keratitis (yellow dots), which is typical for dry eye disease (DED). In the inferior third of the cornea, an epithelial ridge (yellow horizontal line) is visible, indicative of a recently closed erosion. However, the yellow dots and horizontal line are unclear to the audiences. The authors should add some labelling markers to imaging data to identify these important features.
Response to 1.): Thank you very much for your suggestion. We modified the photograph and added a labeling using yellow arrowheads indicating punctate keratitis in a context of DED, and yellow arrows highlighting the epithelial ridge.
- In Section “3. Innovative Antioxidative Approaches for Diabetic Keratopathy”, the authors delve into experimental preclinical findings exploring the efficacy of antioxidative molecules for managing diabetic keratopathy, with the goal of expediting corneal wound healing, alleviating diabetic DED, and mitigating diabetic corneal nerve damage. Nevertheless, in order to allow easy reading and understanding, this reviewer strongly suggests the inclusion of tabular data by summarizing the findings from these literature reports.
Response to 2.): Thank you for your remark. We added a summarizing table at the end of section 3 accordingly.
- As stated by the authors, the disruptions in any component of this unit can result in DED. However, this important claim was not supported by any documented reference. If possible, please refer to the following example paper (DOI: 10.1021/acsnano.3c07817) involving the therapeutic strategy for effective alleviation of DED. The authors are highly recommended to consider the inclusion of this supportive case study in the reference list to update article content and balance scientific viewpoint.
Response to 3.): Thank you for your recommendation. We added a supplement accordingly (lines 315–321, supplementation is underlined).
Reviewer 2 Report
Comments and Suggestions for Authors
the review paper "Diabetic Keratopathy: Redox Signaling Pathways and Therapeutic Prospects" is a very interesting review article reporting peculiar information for keratopathy.
some points must be revised before publication:
a table or figure summarizing the use of compounds with their structures must be useful for the journal.
a selection of new approaches in terms of new drug targets must be included.
Author Response
Reviewer 2
We thank the reviewer for the comments and suggestions. According to these suggestions, we made changes in the text.
- a table or figure summarizing the use of compounds with their structures must be useful for the journal.
Response to 1.): Thank you for your suggestion. We added a table at the end of section 3 accordingly.
- a selection of new approaches in terms of new drug targets must be included.
Response to 2.): Thank you very much for your remark. According to the existing literature, we emphasized, as reported at the end of section 3 (lines 489–491), a selected group of the most promising molecules as antioxidants for diabetic keratopathy such as vitamin D derivatives, α-lipoic acid, and Nrf2 activators such as glycyrrhizin, dimethyl fumarate and mitoquinone.
Reviewer 3 Report
Comments and Suggestions for Authors
The manuscript of Dr. Buonfiglio and colleagues summarizes the pathophysiology of diabetic keratopathy and examines in detail the role of oxidative stress. This review is well-organized and exhaustive and expands a previous, although recent, contribution of the same Authors (doi.org/10.1016/j.redox.2023.102967). I only have a few suggestions that the Authors may consider.
Lines 70-99: in paragraph 2.1.1 there are some abbreviations that are never used in the rest of the manuscript. Please revise all the abbreviations.
Lines 115-116: since refs 45-49 are not specifically concerned with keratopathy or, in general, with corneal disease, it should be better explained how the Authors can state that these processes induce cell death in keratocytes.
Fig. 2: Not all abbreviations are explained in the figure legend (for example, VEGF-A and VCAM are missing). In addition, I would suggest to organize the abbreviations in alphabetical order (also in fig 4 and 5).
Lines 185-186: While the function of neurotrophins in the cornea are explained in the following lines, no details are given on the function of neuropeptides or of the autonomic innervation.
Chapter 3: A variety of antioxidant compounds are described here as potential therapeutic agents for diabetic keratopathy. Although one can imagine that, being the cornea the target organ, these agents are applied topically (eye drops?), only in a couple of cases topical administrations are specified, suggesting other types of applications in other cases. Please clarify the routes of drug administration in the in vivo experiments. In addition, I suggest to clearly separate the description of the in vitro and the in vivo observations in each paragraph.
Author Response
The manuscript of Dr. Buonfiglio and colleagues summarizes the pathophysiology of diabetic keratopathy and examines in detail the role of oxidative stress. This review is well-organized and exhaustive and expands a previous, although recent, contribution of the same Authors (doi.org/10.1016/j.redox.2023.102967). I only have a few suggestions that the Authors may consider.
- Lines 70-99: in paragraph 2.1.1 there are some abbreviations that are never used in the rest of the manuscript. Please revise all the abbreviations.
Response to 1.): Thank you very much for your recommendation. We deleted some abbreviations that we later do not use in the text. We revised all the abbreviations in the whole text accordingly.
- Lines 115-116: since refs 45-49 are not specifically concerned with keratopathy or, in general, with corneal disease, it should be better explained how the Authors can state that these processes induce cell death in keratocytes.
Response to 2.): Thank you for your remark. Due to investigations conducted on diabetic corneas that specifically demonstrated the role of pyroptosis in corneal epithelia and in nerves (for example ref. 44), and which revealed a beneficial effect in suppressing the pyroptotic pathway (refs. 151; 152; 154; 155), we underline the role of pyroptosis in diabetic keratopathy. Through refs. 45–49, we do explain how pyroptosis leads to cell death.
- 2: Not all abbreviations are explained in the figure legend (for example, VEGF-A and VCAM are missing). In addition, I would suggest to organize the abbreviations in alphabetical order (also in fig 4 and 5).
Response to 3.): Thank you very much for your suggestion. We revised all the abbreviations in the figures 2, 4 and 5, and we organized them in alphabetical order.
- Lines 185-186: While the function of neurotrophins in the cornea are explained in the following lines, no details are given on the function of neuropeptides or of the autonomic innervation.
Response to 4.): Thank you for your remark. We added a supplement on neuropeptides (lines 201–211, section underlined).
- Chapter 3: A variety of antioxidant compounds are described here as potential therapeutic agents for diabetic keratopathy. Although one can imagine that, being the cornea the target organ, these agents are applied topically (eye drops?), only in a couple of cases topical administrations are specified, suggesting other types of applications in other cases. Please clarify the routes of drug administration in the in vivo experiments. In addition, I suggest to clearly separate the description of the in vitro and the in vivo observations in each paragraph.
Response to 5.): Thank you very much for your suggestion. We now report on the route of administration in all the described in vivo studies and we further clearly specify the description in vitro and in vivo in each paragraph of section 3 (changes are underlined). Moreover, we have organized the findings of all the reported studies in a table at the end of section 3.
Reviewer 4 Report
Comments and Suggestions for Authors
The present review is intriguing. The authors described in detail all molecular mechanisms and aspects of the clinical entity of Diabetic keratopathy and its treatment.
Is keratopathy a frequent complication of DM or a rare one?
Which cell types have been implicated in the pathogenesis of keratopathy? Some cells need insulin to use glucose, and others do not. The last one is directly exposed to the harmful effects of hyperglycemia. The authors should discuss which cells are mainly involved in the pathogenesis of keratopathy.
Moreover, what is the role of topical insulin use in the treatment of keratopathy? Insulin is a growth factor that can promote corneal epithelial cell proliferation and migration. In addition, it can also inhibit corneal epithelial cell apoptosis. Which is the molecular mechanism?
The methodology is appropriate. The manuscript is well written, and the discussion/conclusions are acceptable.
Overall, data are of interest.
Comments on the Quality of English Languagenone
Author Response
The present review is intriguing. The authors described in detail all molecular mechanisms and aspects of the clinical entity of Diabetic keratopathy and its treatment.
1.) Is keratopathy a frequent complication of DM or a rare one?
Response to 1.) Thank you for your question. As reported by lines 44–45, according to the cited articles (refs 12–17), diabetic keratopathy can affect over the half of the diabetic population. Hence, although widely underdiagnosed and overlooked, diabetic keratopathy can be considered a frequent complication of DM. We added a statement accordingly (lines 45–46, changes are underlined).
2.) Which cell types have been implicated in the pathogenesis of keratopathy? Some cells need insulin to use glucose, and others do not. The last one is directly exposed to the harmful effects of hyperglycemia. The authors should discuss which cells are mainly involved in the pathogenesis of keratopathy.
Response to 2.) Thank you very much for your remark. We added a passage and references accordingly (lines 67–77, changes are underlined).
3.) Moreover, what is the role of topical insulin use in the treatment of keratopathy? Insulin is a growth factor that can promote corneal epithelial cell proliferation and migration. In addition, it can also inhibit corneal epithelial cell apoptosis. Which is the molecular mechanism?
Response to 3.) Thank you very much for your question. We added a passage on that issue and references accordingly (lines 460–478, changes are underlined).
Round 2
Reviewer 4 Report
Comments and Suggestions for Authors
no other comments
Comments on the Quality of English Languagenone